# LLM-based SQL Generation with Reinforcement Learning

**Mariia Berdnyk[1], Marine Collery [1]**

[1]IBM France Lab
mariia.berdnyk@ibm.com, marine.collery@ibm.com

## Abstract

The text-to-SQL problem remains a challenging task, even with the advancements of Large Language Models (LLMs). Current state-of-the-art models require extensive preprocessing steps and powerful LLMs to achieve accurate SQL query generation, which leads to significant resource utilization. We introduce two models deriving from one another SQL-RL-GEN and SQL-RL-GEN*, that improve text-to-sql generation while minimizing the resources needed for training and maximizing flexibility. The SQL-RL-GEN generates a reward function to guide the agent's training process, while SQL-RL-GEN* uses this reward function to tune a base LLM in solving the specified task. Our models achieve an accuracy improvement of 2-7% compared to state-of-the-art methods on a limited training dataset composed of only 1000 samples and with a small LLM of 248M parameters.

**Code** — https://github.com/IBM/sql-rl-gen
**Datasets** — https://ibm.box.com/v/sql-rl-gen-data

## Introduction

Large Language Models (LLMs) have exhibited remarkable capabilities in various tasks, including text and code generation problems (Jiang et al. 2024). The success is largely attributed to the vast amount of data available for training and tuning processes.

The text-to-SQL generation problem is a critical area of research within the fields of natural language processing (NLP) and database systems. Since SQL remains one of the most widely used programming languages for database management (51.52%), the text-to-SQL translation enables non-skilled users to access structured databases like engineers using everyday language (Hong et al. 2024).

Current text-to-SQL best models, which achieve the top scores on the most comprehensive SQL datasets, are based on modifying the model structure by providing several other preprocessing steps in between the model and SQL generation. For instance, `ExSL + granite-34b-code` by IBM Research combines 2 steps before passing the question to the model, which are: schema linking and content linking (Martineau 2024). SQLNet uses a sketch-based approach,

incorporating a dependency graph to guide token predictions based on their dependencies (Xu, Liu, and Song 2017). However, the question remains open whether generations without a solid data background can be further improved and generalized easily, no matter the model used. Another approach based on Reinforcement Learning (RL), Seq2SQL, uses basic rewards (1 for correct query generation and -1 otherwise) obtained from in-the-loop query execution over the database to learn a policy for generating the better query (Zhong, Xiong, and Socher 2017). Despite the impressive results that Seq2SQL has demonstrated at the time of its publication, subsequent work suggest that the base reward is not enough to solve the text-to-SQL problem (Xu, Liu, and Song 2017).

Reward function design for generation task demands significant human effort and is known to be notoriously difficult in practice (Sutton and Barto 1995). For this purpose, recently, a generic novel reward design algorithm, EUREKA (Ma et al. 2024), powered by coding LLMs was proposed. Unlike prior works using LLMs to aid reward design, EUREKA is completely free of task-specific prompts, reward templates, as well as few-shot examples (Ma et al. 2024). Instead, it uses evolutionary search and feedback to generate the best reward function with LLM.

In this paper, we introduce two models deriving from one another SQL-RL-GEN and SQL-RL-GEN*.

**SQL-RL-GEN** algorithm finds the best reward function (reference reward function) to be used for the training of an RL agent to generate SQL queries from text with similar techniques as proposed by EUREKA i.e. implementing the reward design for SQL generation, feedback formulation and an evolutionary search of the best reward function.

**SQL-RL-GEN\*** uses the reference reward function generated by SQL-RL-GEN on a reference dataset to tune a base LLM (`flan-t5-base`) for SQL generation with limited resources.

The approach makes the following key contributions compared to existing work:

1. **Versatility and efficiency of the reference reward function for SQL generation**: SQL-RL-GEN* outperforms state-of-the-art SQL generation models on a different dataset than the one used to generate the reference reward function, with only 1000 samples used for training and a relatively small base LLM of 248M parameters.

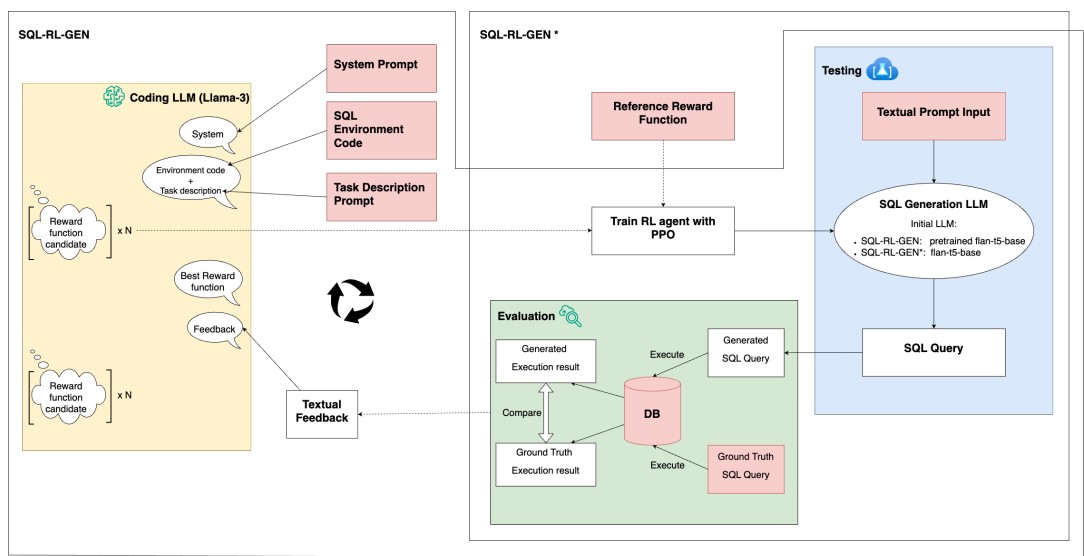

Figure 1: SQL-RL-GEN takes as inputs: a system prompt, an SQL environment code, and a task description prompt. The coding LLM iteratively generates N reward function candidates, each used to train an SQL generation model from scratch with the RL Proximal Policy Optimization (PPO) algorithm. The resulting models are evaluated by comparing the rows obtained from generated SQL queries execution with those from ground truth queries. The evaluation results (feedback) and the best selected by accuracy reward function are fed back to the coding LLM for the next iteration. SQL-RL-GEN* is a special case where the best reward function from a previous SQL-RL-GEN training is used directly to train the RL agent.

This makes SQL-RL-GEN* efficient in terms of resource utilization.

2. **Domain adaptability**: SQL-RL-GEN algorithm is easily adaptable for generating reward functions in various text-to-code domains, enabling its application in diverse settings.

## Problem Statement

Given a textual prompt input $p$, which is the part of the set of all possible textual prompts $\mathcal{P} = \{p_1, p_2, ..., p_n\}$, and an LLM $L\colon \mathcal{P} \to \mathcal{O}$ that maps prompts to code outputs in the space of all possible code outputs $\mathcal{O} = \{o_1, o_2, ..., o_m\}$, our goal is to train $L$ to generate an SQL query $s \in \mathcal{S}$ from the input prompt $p$, where $\mathcal{S} \in \mathcal{O}$ is the set of all possible SQL queries.

The prompt is represented as $p = (I, T, Q)$, where:

- $I$ is a set of possible instructions, e.g., "convert", "summarize", "answer", etc. It can be represented as a binary vector $i \in \{0, 1\}^{|I|}$, where each element corresponds to one of the instructions in $I$.
- $T$ is a set of possible table schemas: $T = (t_1, t_2, ..., t_j)$. $t$ is a single table, represented as a tuple of columns $t = (c_1, c_2, ..., c_k)$ where $k$ is the number of columns in the table $t$.
- $Q$ is a set of possible questions, e.g., "How many...", "What is...", etc. Each question can be represented as a string $q$.

As instruction ($I$) for the problem remains unchanged, training and testing datasets consist of pairs of input data $(t, q)$ and corresponding (ground truth) query $s$ such that a dataset $D$ is defined as $D = ((t_1, q_1), s_1), ...((t_N, q_N), s_N)$ where $N$ is the number of samples.

Once trained, model $L_{trained}$ should return for a specific prompt $p$ a generated SQL query $s_{gen}$ to be compared with corresponding (ground truth) query $s$.

## Method

An overview of the approach of SQL-RL-GEN is illustrated in Figure 1. An initialization step is followed by a loop composed of:

- the generation of a reward function,
- the training of the RL agent,
- the evaluation of the tuned SQL generation model and the supply of textual feedback.

**Initialization.** In the initialization stage, similarly to EUREKA original approach, we provide the LLM with a prompt that outlines the task and SQL environment. It is composed of the following parts.

1. The **system prompt** explicitly defines the role of the LLM as a reward engineer and provides an example of the reward function signature.

2. The **task description** specifies the goal of the model during training and generation. For SQL generation, it is set to "Converting question and database tables into SQL query".

3. The **SQL environment** component is crucial and provides the LLM with context where the trained agent will operate and execute generated reward functions during training. In the same manner as in EUREKA, SQL-RL-GEN feeds the raw environment source code (excluding

reward code, if present) as context with minimal explanations of external functions (Ma et al. 2024).

The entire initialization stage sets the generation goal, allowing adaptation to different tasks by modifying the initial prompts to solve similar problems in a comparable manner. All initialization prompts are available in Appendix.

**Reward Function Generation and Training.** Thanks to the provided prompts, the coding LLM generates multiple reward functions that are used to train RL agents with PPO (Schulman et al. 2017) algorithm in a similar manner to EU-REKA, and obtain a tuned SQL generation LLM.

**Evaluation and Feedback.** In order to improve the next iteration of reward function generation, textual feedback on the performance of the best tuned SQL generation LLM is provided to the coding LLM as well as the reward function, with which this model is trained. The SQL generation LLM is considered the best (out of the multiple generated), if after training it yields higher average accuracy during the evaluation step than other models from both previous and current iterations.

To evaluate the tuned SQL generation LLM performance, similarly to Seq2SQL approach, both SQL-RL-GEN and SQL-RL-GEN* evaluation step consists in comparing the SQL rows resulting from the execution of the generated SQL query and the ones obtained with the ground truth query. The generated queries are only executed when they do not modify the execution environment.

The evaluation results are saved, converted into text and provided back to the LLM as feedback with quantitative information of the performance (accuracy, precision, recall, F1-score and intersect over union (IoU)). In addition, if errors are encountered during the execution of generated queries, error types along with the error descriptions are returned in the feedback. The error descriptions do not provide specific information about the database context and are data independent.

As shown in Figure 1, SQL-RL-GEN* is derived from SQL-RL-GEN and consists in retrieving the best generated reward function from a former training of SQL-RL-GEN and using it to directly train a RL agent.

## Experiments

In order to evaluate the validity and usefulness of SQL-RL-GEN, we apply it on Spider dataset (Yu et al. 2019) to obtain our reference reward function. The WikiSQL dataset (Zhong, Xiong, and Socher 2017) is then used to evaluate the validity and robustness of this reference reward function, SQL-RL-GEN*.

**Spider Dataset** Spider consists of 10181 questions and 5693 unique complex SQL queries on 200 databases with multiple tables covering 138 different domains. In Spider 1.0, different complex SQL queries and databases appear in train (8659 examples) and test (1034 examples) sets.

**WikiSQL Dataset** WikiSQL consists of a corpus of 87726 hand-annotated SQL query and natural language question pairs. These SQL queries are further split into training

(61297 examples), development (9145 examples) and test sets (17284 examples).

**Experimental Setting.** For each dataset, a subset of 1000 randomly selected samples are used for training and another subset of 1000 randomly selected samples are used for testing. The experiments are carried out with k-fold cross-validation strategy with k = 5.

The reward function generation and reflection are implemented using `llama-3-405b-instruct` (Touvron et al. 2023). This model is free, open-source and is known for its good instructed generation capabilities (Touvron et al. 2023), which makes it the better choice than the proprietary one described in the EUREKA reference paper. Characteristics of the model are available in Appendix, Table 4. The initial LLMs (agents) used for generating SQL queries are `flan-t5-base` (Chung et al. 2024) and a pretrained version of `flan-t5-base` on SQL syntax (noa 2023). `flan-t5-base` transformer-based model consists of only 248 million parameters, which makes its training process computationally efficient and light. To evaluate the efficiency of SQL-RL-GEN*, trained `flan-t5-base` was compared with the trained on the same samples Seq2SQL and SQLNet reference models, which are configured according to their original papers. All agents characteristics can be found in Appendix, Table 5.

PPO algorithm is configured in the exact same manner as in (Schulman et al. 2017) and as described in EUREKA reference paper. The parameters are listed in Table 6 in Appendix. However, unlike the original PPO approach, which only allows a single trial per sample before switching to another, for the training of SQL-RL-GEN and SQL-RL-GEN*, we introduce an improvement by enabling the model to experiment 10 times on the same sample before moving on. This approach enables the agent to learn from its mistakes and refine its policy for generating better SQL queries. By allowing multiple trials on the same sample, we can more effectively capture the nuances of text generation problems, which often demand a more refined approach than the original single-trial method. This modification allows our model to learn from its errors and improve the quality of subsequent SQL generations.

All experiments are GPU-based and were conducted on a Lenovo ThinkPad P15 Gen 1 with Intel Core i7-10750H CPU, 12 Cores, Quadro T1000/PCIe/SSE2 graphics with 4Gb of memory and running Red Hat Enterprise Linux 8.10.

## Preliminary Results

**SQL-RL-GEN and Reference Reward Function Generation.** Training SQL-RL-GEN on Spider dataset, with `flan-t5-base` model as initial SQL generation LLM, does not lead to any improvements in terms of accuracy ($O\%$). This is due to the fact that the `flan-t5-base` model has not been trained on any code or SQL queries, and that the training on Spider dataset is severely limited by the constrained size of 1000 training samples and that Spider features highly intricate and complex queries. However, as shown in Table 1 when training SQL-RL-GEN on Spider dataset, with a pretrained for SQL syntax `flan-t5-base`

|  | pretrained flan-t5-base | SQL-RL-GEN |
|---|---|---|
| accuracy (%) | $44.7 \pm 1.6$ | $48.0 \pm 0.78$ |
| exec $s_{gen}$ (%) | $61.5 \pm 1.5$ | $64.3 \pm 1.3$ |

Table 1: Average accuracies and percentages of generated executable queries $s_{gen}$ along with standard errors for 5-fold cross validation for the initial LLM (`flan-t5-base` pretrained on SQL syntax model) and after SQL-RL-GEN training on Spider dataset. Metrics shown are obtained on Spider testing dataset.

|  | Seq2SQL | SQLNet | SQL-RL-GEN* |
|---|---|---|---|
| accuracy | 7.1% | 11.3% | **13.8%** |
| exec $s_{gen}$ | 12.8% | 12.1% | **30.6%** |

Table 2: Accuracies and percentages of executable generated queries $s_{gen}$ for Seq2SQL, SQLNet and SQL-RL-GEN* obtained on WikiSQL test dataset.

model as initial SQL generation LLM, the performance in terms of accuracy is improved by more than $3\%$ and on average there are almost $3\%$ more executable generated queries.

**Versatility of the Reference Reward Function.** As shown in Table 2, SQL-RL-GEN* which uses the reference reward function to fine-tune `flan-t5-base`, outperforms state-of-the-art models Seq2SQL (Zhong, Xiong, and Socher 2017) and SQLNet (Xu, Liu, and Song 2017) on WikiSQL dataset both in terms of accuracy and number of executable generated SQL queries. It points out the versatility of the reference reward function and how efficient in terms of resource utilization SQL-RL-GEN* is, as only 1000 samples were used for training compared to the entire dataset for the other models.

**Reusability of the Reference Reward Function.** Finally, in order to validate that the reference reward function can also be used in other RL-based algorithm, we compared Seq2SQL model to a version of Seq2SQL trained with our reference reward function version as shown in Table 3. The metrics employed for model evaluation align with those utilized in the Seq2SQL original paper (and are described in Appendix). Again, usage of the reference reward function improved all of the different accuracies defined in (Zhong, Xiong, and Socher 2017) to evaluate SQL generation. This reward function can therefore be reused in other RL-based context in the text-to-SQL generation field.

## Limitations and Future Directions

While SQL-RL-GEN and SQL-RL-GEN* show strong improvements with limited data, further analysis is needed:

1. **Error Mitigation**: The reward function penalizes syntax errors, logical inconsistencies, and schema mismatches. A detailed breakdown of its impact on correction rates would clarify its role in improving performance.

|  | Seq2SQL | Seq2SQL with SQL-RL-GEN* reference reward function |
|---|---|---|
| Dev Acc$_{qm}$ | 53.1% | **55.0%** |
| Dev Acc$_{exec}$ | 60.4% | **62.5%** |
| Test Acc$_{qm}$ | 52.7% | **55.3%** |
| Test Acc$_{exec}$ | 60.0% | **63.2%** |

Table 3: Accuracy comparison on WikiSQL dataset between Seq2SQL and Seq2SQL with SQL-RL-GEN* reference reward function. Acc$_{qm}$ and $Acc_{exec}$ indicate the query-match (string match) and the execution accuracy (correct result) (Zhong, Xiong, and Socher 2017) respectively on development and testing datasets.

2. **Generalization**: The model improves when transferring from Spider to WikiSQL, but its adaptability to unseen schemas requires further evaluation across diverse benchmarks.

3. **PPO Trials**: Additional trials refine the reward function but increase computational cost. Analyzing diminishing returns could optimize efficiency.

4. **Scalability**: Testing on varied datasets and resource constraints would help assess robustness and adaptability.

## Conclusion

We have presented SQL-RL-GEN and SQL-RL-GEN* deriving from one another. The first one proposes a reference reward function calibrated for SQL generation thanks to evolutionary search and feedback formulation (Ma et al. 2024) that can be used by the second to tune LLM with limited resources. The experiments demonstrated that SQL-RL-GEN* outperforms state-of-the-art methods and that the reference reward function can boosts the generation capability of RL-based methods on WikiSQL and Spider datasets.

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

## Appendix - Initialization Prompts

### System Prompt

```
You are a reward engineer trying
to write reward functions to solve
reinforcement learning tasks as
effective as possible. Your goal
is to write a reward function for
the environment that will help the
agent learn the task described in
text. Your reward function should use
useful variables from the environment
as inputs. An example of the reward
function signature can be:
    ```python
        {task_reward_signature_string}
    ```

  You need to generate the reward
functions of EXACTLY this syntax.
Everything else is not accepted. Please
make sure that the code is compatible
with Gym env. **PROVIDE ONLY PYTHON
CODE.**
```

### Task Description

```
The Python environment is
{task_environment_code_string}. Write a
reward function for the following task:
{task_description}.
```

## SQL environment

```python
class SQLRLEnv(TextRLEnv):
    def __init__(self, model,
        tokenizer, dataset, ...):
        super().__init__(model,
            tokenizer,
            observation_input,
            max_length,
            compare_sample,
            unfreeze_layer_from_past)
        ...
    def sql_query_execution_feedback
        (self, input_item,
        predicted_text) -> Dict:
        ...

    # Base method
    def get_reward(self, input_item,
        predicted_list, finish):
        if finish:
            predicted_text = self.
                tokenizer.
                convert_tokens_to_string
                (predicted_list[0])
            reward, metrics = self.
                compute_reward(
                input_item,
                predicted_text)
            metrics["reward"] =
                reward
            ...
            return reward
        return 0.0

    # Skeleton of generation
    def compute_reward(self,
        input_item, predicted_text)
        -> Tuple[float, Dict]
```

## Appendix - Experimental Settings

|                      | llama-3-405b -instruct |
| --- | --- |
| Number of parameters | 405B |
| Temperature | 0.95 |
| Context size | 15 000 |
| Decoding method | sample |

Table 4: llama-3-405b-instruct and flan-t5-base characteristics.

| | flan-t5-base | SQLNet | Seq2SQL |
|---|---|---|---|
| Architecture | Encoder-Decoder Transfomer (T5) | BiLSTM + attention + seq2set | Encoder-Decoder + RL |
| Number of parameters | 248M | 38.5M | 37M |
| Pretrained | Yes | No | No |
| Fine-tuning required | Yes | Yes | Yes |
| Temperature | 0.8 | 0.8 | 0.8 |

Table 5: Experimental flan-t5-base, Seq2SQL and SQLNet agents models characteristics.

| Parameters | Values |
|---|---|
| Tensors type | F32 |
| Temperature | 0.8 |
| Top k | 100 |
| Top p | 0.85 |
| Update interval | 50 |
| Minibatch size | 512 |
| Number of Epochs | 5000 |
| Number of steps | 1000 |
| Number of evaluation episodes | 5 |
| Maximum training episodes length | 1000 |
| Evaluation interval | 10 |
| Maximum new tokens | 250 |
| Minimum new tokens | 10 |

Table 6: PPO algorithm settings.

# Appendix - Reference Reward Function generated with SQL-RL-GEN

```python
def compute_reward(self, input_item, predicted_text) -> Tuple[float, Dict]:
    feedback = self.sql_query_execution_feedback(input_item, predicted_text)
    if "error_reason" in feedback and feedback["error_reason"] is not None:
        if feedback["not_sql_format"]:
            return -100.0, feedback
        elif "no such column" in feedback["error_reason"]:
            return missed_key_words(-3.0, feedback["expected"], predicted_text), feedback
        else:
            return missed_key_words(-50.0, feedback["expected"], predicted_text), feedback
    # Calculate a weighted average of the metrics
    accuracy_weight = 3
    precision_weight = 2
    recall_weight = 2
    f1_weight = 3
    if "accuracy" in feedback:
        reward_accuracy = accuracy_weight * feedback["accuracy"]
    else:
        reward_accuracy = 0
    if "precision" in feedback:
        reward_precision = precision_weight * feedback["precision"]
    else:
        reward_precision = 0
    if "recall" in feedback:
        reward_recall = recall_weight * feedback["recall"]
    else:
        reward_recall = 0
    if "f1" in feedback:
        reward_f1 = f1_weight * feedback["f1"]
    else:
        reward_f1 = 0
    reward = reward_accuracy + reward_precision + reward_recall + reward_f1
    if reward == 10.0:
        return reward, feedback
    return missed_key_words(reward, feedback["expected"], predicted_text), feedback
```

Figure 2: Reference Reward Function generated with SQL-RL-GEN and used for training of SQL-RL-GEN*.