# OpenReview forum: "LLM-based SQL Generation with Reinforcement Learning"
_AAAI.org/2025/Workshop/NeurMAD — AAAI 2025 Workshop NeurMAD Submission_

### Official Review · Reviewer_bTRR · 2024-12-15
**Good application of EUREAK PPO for text-to-SQL task with good experiments and results**

**Rating:** 7
**Confidence:** 5

**Review:**

This paper proposes a RL-based approach to fine tune a relatively small LM with limited samples (~1000) to outperform the SOTA  for the task of text-to-sql. The paper was well written and the experiment and preliminary results seem to be good.

Here are some strengths of this paper:
1. Relevance of this paper’s topic is high: The paper picks a relatively difficult problem: text-to-SQL, which has big impact if such task can be solved to the accuracy of human-level performance (accuracy for SQL engineers is more than 93% according to IBM text-to-SQL generator study). It is of great interest to many industry practitioners as well.
2. Applying a modified EUREKA PPO algorithm for reward function generation: this paper uses the EUREKA PPO with multiple trials on the same sample to generate reward function. EUREKA is a gradient-free, evolution search-based algorithm, which performs well without any reward templates etc.
3. Evaluation of the experiments is good: this paper uses 1 LLM for reward function generation, and then uses another smaller LLM for fine tuning using the reward function generated by the previous step . These two steps then iterate until reaching some stopping criteria. The results seem solid, although improvement comparing with the previously published results is not that significant, but still reasonably better!

Some weakness of this paper:
1. Better clarity of presentation: The paper refers EUREKA at the introduction section and the SQL environment setup, but did not mention EUREKA any more in the experiments section when it describes the PPO . It would be better to describe how the EUREKA configuration is different from the PPO reference it cites. Also, it would be better to describe EUREKA and PPO in the background/introduction section together and then compare if they would use one vs another in a more clear way.
2. Choices of LLMs to the impact of the approaches: since this paper only uses 1 LLM llama-3-405b-instruct for reward generation and 1 LLM flan-t5-base for fine tuning using the generated reward function, it is not clear whether the choices of LLM will matter. In particular, the latest OpenAI GPT-4o vs. other SOTA code optimized pertained LLMs are not being evaluated . The baseline model chosen for fine tuning is also not necessarily representative and it is not clear. So it is not clear whether the choice of these LLMs in this paper is arbitrary, hence it is hard to tell whether the RL approach is really impactful enough if some other LLMs are used.
3. Choices of sample size for fine tuning is not well justified: while the paper tries to suggest that they use a small size of samples (1000 samples) , it is not clear how they come up this size. I would like to see some scaling experiments where they can change the size of fine tuning samples from some smaller samples and then scale to larger size: for example, from 100 samples, to 500, 1000, 5000, 10,000, etc, to see the impact of the size of the samples on the final performance of the SQL generation fine-tuned model.
4. DPO vs. PPO choice: it would be interesting to see if the paper can also address if DPO is applicable for this use case, where human preference can be directly optimized/fine tuned. That will change the approach but it is worthy clarifying , at least in the background section.

Overall, this paper seems to have some good ideas and experiments with a few weaknesses, but it is good enough for the workshop to accept this for presentation and discussion.

---

### Official Review · Reviewer_RFLD · 2024-12-25
**a novel method targeting at wide industrial application**

**Rating:** 6
**Confidence:** 3

**Review:**

This paper proposed a novel reinforcement learning method for solving the task of generating SQL statements from questions in natural languages. Authors introduce two models: (1) SQL-RL-Gen model generates a reward function. This function improves the training process. (2) SQL-RL-Gen* model uses the generated reward function to tune the LLM model. Experimented with a limited amount of training data, this new method achieved better performance than state-of-the-art methods.

Questions:
1) Is it possible to enumerate "all possible text prompts" in real applications?
2) It seems that authors used (Schulman et al 2017)'s method to generate reward functions. Is the method too old? What is the limitation of the method?
3) In experiments, authors repeated 10 times on the same sample before moving on. Did (how often) LLM ignore feedbacks?

---

### Official Review · Reviewer_ks6w · 2024-12-28
**Novel Approach on Improved Text 2 SQL with reduced param model, Can Improve paper quality with performance metrics and details on reward generation**

**Rating:** 6
**Confidence:** 4

**Review:**

## Summary
---
This paper introduces SQL-RL-GEN and SQL-RL-GEN*, two novel approaches for text-to-SQL generation using reinforcement learning and large language models. The work addresses the challenge of generating SQL queries from natural language while minimizing computational resources. The authors propose using a reference reward function generated by SQL-RL-GEN to guide the training process, which is then utilized by SQL-RL-GEN* to fine-tune a base LLM. The paper demonstrates improved performance using only 1,000 training samples and a relatively small 248M parameter model.

## Strengths
---
- Strong empirical results showing improved accuracy (2-7%) over state-of-the-art methods while using only 1,000 training samples
- Resource-efficient approach that achieves good performance with a small base model while demonstrating versatility across different datasets

## Suggestions for Improvements
---
- The paper lacks a comprehensive analysis of model parameter counts and computational requirements compared to baseline methods like SQLNet and Seq2SQL, making it difficult to fully assess efficiency claims
- A more thorough comparison with other reward-based approaches in text-to-SQL generation would strengthen the paper's contribution
- More detailed analysis of failure cases and limitations would help guide future research in this direction

---

### Decision · Program_Chairs · 2024-12-30

**Decision:**

Accept

**Comment:**

 This is a work that fits into the scope of industrial applications.